# The Blood–Brain Barrier and Pharmacokinetic/Pharmacodynamic Optimization of Antibiotics for the Treatment of Central Nervous System Infections in Adults

**DOI:** 10.3390/antibiotics11121843

**Published:** 2022-12-19

**Authors:** Nicholas Haddad, Maddie Carr, Steve Balian, James Lannin, Yuri Kim, Courtney Toth, Jennifer Jarvis

**Affiliations:** 1College of Medicine, Central Michigan University (CMU), Mt Pleasant, MI 48859, USA; 2Covenant HealthCare, Saginaw, MI 48602, USA; 3CMU Medical Education Partners, Saginaw, MI 48602, USA; 4Ascension St. Mary’s Hospital, Saginaw, MI 48601, USA

**Keywords:** pharmacokinetics, pharmacodynamics, central nervous system infections, meningitis, ventriculitis, brain abscess, blood–brain barrier, antibiotic, antimicrobial, the specific classes and names of antimicrobial agents discussed in the review

## Abstract

Bacterial central nervous system (CNS) infections are serious and carry significant morbidity and mortality. They encompass many syndromes, the most common being meningitis, which may occur spontaneously or as a consequence of neurosurgical procedures. Many classes of antimicrobials are in clinical use for therapy of CNS infections, some with established roles and indications, others with experimental reporting based on case studies or small series. This review delves into the specifics of the commonly utilized antibacterial agents, updating their therapeutic use in CNS infections from the pharmacokinetic and pharmacodynamic perspectives, with a focus on the optimization of dosing and route of administration that have been described to achieve good clinical outcomes. We also provide a concise synopsis regarding the most focused, clinically relevant information as pertains to each class and subclass of antimicrobial therapeutics. CNS infection morbidity and mortality remain high, and aggressive management is critical in ensuring favorable patient outcomes while averting toxicity and upholding patient safety.

## 1. Introduction

CNS infections are serious and carry significant morbidity and mortality, oftentimes with devastating outcomes. In a recent retrospective review by Sunwoo et al. of confirmed meningitis in patients admitted between 2007 and 2016, the in-hospital mortality was 10.6%, and 3 months after discharge it was 14.8%, with significant neurological complications in 39.1% of patients [1]. Overall mortality was reported even higher, at 21%, from a cohort of patients in the Netherlands in 2004 by De Beek et al., with different rates of mortality associated with different meningitis syndromes, and an unfavorable outcome in 34% of all cases [2]. CNS infections require immediate and aggressive management, with antimicrobial agents targeted against the most likely organism and subsequently appropriately tailored based on culture and non-culture data. Delay or lack of prompt therapy results in higher mortality [3,4]. Antimicrobial drug levels in the CSF are completely dependent on penetration from serum, as they are not metabolized in the CSF. Exit of drugs from the CSF is managed by the choroid plexus via energy-dependent pumps, which transport molecules one way back to serum [5]. Ensuring an adequate drug level at the site of infection, the CNS, is crucial in achieving cure but challenged by the presence of the blood–brain barrier (BBB) and blood–cerebrospinal fluid barrier (BCSFB). The intrinsic role of those barriers, very similar physiologically, is primarily to protect the brain and spinal cord from compounds in the general circulatory system, a phenomenon first described in the 1880s by Paul Ehrlich [6,7]. For this review, discussion of the BBB also includes roles of the BCSFB. The BBB is composed of specialized endothelial cells with intercellular tight junctions and increased numbers of pinocytotic vesicles in microvascular endothelial cells [8] that reduce access of bloodborne compounds into the CSF. Meningeal damage from inflammation in meningitis disrupts this mechanism, facilitating the entry of molecules from the serum to the CSF [9]. The use of corticosteroids in inflammatory disorders, such as meningitis, reduces inflammation and consequently drug entry into the CSF, although this has not been shown with vancomycin in some studies [10,11,12]. Steroid use in CNS infections is not a focus of this review, but it is still noteworthy to mention that steroids are indicated in certain CNS infections but not in others [13], which indirectly suggests that the reduction in antimicrobial transfer across the BBB in inflamed meninges is one of many biological parameters in this complex pharmacokinetic formula. Likewise, entry of an antimicrobial drug from blood to the CSF is facilitated by other intrinsic properties related to the drug itself. It is more efficient in compounds with low molecular weights, a lower ionization degree at physiologic pH, high lipid solubility (lipophilicity), and low degree protein binding [14,15].

Pharmacodynamics (PD) is the study examining the effect of drugs on the human body, as pertains to the time and concentration of antimicrobials at the site of infection, the CNS in this review. Pharmacodynamics define dosing and administration frequency, the goal of which is ensuring optimal efficacy of the antimicrobial agent at the site of infection. Pharmacodynamically, beta-lactams exhibit time-dependent activity, such that the time of the free (non-protein bound) drug exposure above the MIC is the major determinant of activity (fT > MIC), and higher drug levels would not cause more killing of microorganisms [11,14,16,17,18,19]. Hence, the goal of dosing those agents is always ensuring a drug level above the MIC during use, preferably four times the MIC of the targeted organism [20,21]. On the other hand, most other classes exhibit concentration-dependent activity either solely or with some degree of time dependence as well. For example, aminoglycosides, rifamycins, and fluoroquinolones cause maximal killing when their concentrations are maximized (AUC/MIC), even when serum levels eventually fall below the MIC (Cmax/MIC). Hence, the goal of dosing those agents is ensuring such high concentrations over a snapshot of time of use. Part of the efficacy for this mechanism is the ability of those agents to exert a post-antibiotic effect (PAE) defined by the stunting of bacterial regrowth after the levels in serum fall below their MIC.

On the other hand, pharmacokinetics (PK) describes the processes that govern the passage of the different drugs throughout the human body, which results in different concentrations in different body compartments [11,14,17,18,22,23]. Specifically, absorption, distribution, metabolism, and excretion are the major parameters that define PK, and its clinical application is ensuring the safe and effective therapeutic delivery of drugs to where their action is needed. The most reliable measure of CSF penetration of a drug from serum is the AUC_CSF_/AUC_plasma_ ratio [24,25]. This metric will be described for many of the antimicrobials discussed here.

This review article is an update and summary of the literature that has analyzed the PK/PD properties of antibiotics used in CNS infections, with a specific objective of discussing optimization tools to achieve a successful therapeutic target that results in favorable clinical outcomes. We reviewed MEDLINE/PubMed and Google Scholar publications, between 1985 and July 2022. Additional papers were extracted from the references of retrieved articles based on the clinical relevance of the specific perspective being reviewed. For a detailed explanation of the PK/PD metrics, the reader is referred to Appendix A.

The final number of unique publications reviewed was 213.

The choice of antimicrobials in the management of CNS infections depends on several factors, the most important and immediate of which is the empiric selection of an agent that targets the likely organism(s) and susceptibilities. Another factor is the degree of penetration into the CSF via the BBB (and BCSFB), which is in turn defined by multiple host and antimicrobial drug-related characteristics: the presence or absence of inflammation, the molecular structure of the antimicrobial (e.g., hydrophobicity/lipophilicity), protein-binding characteristics [15], its molecular weight, the degree of renal clearance, and the rate of CSF production, which is oftentimes enhanced during inflammation [26]. Upon entry into the CSF, the antimicrobial should ideally have a high and rapid degree of bactericidal activity. The most common route of administration is intravenous (IV), although direct administration into the CSF, either intraventricularly (intraventricular therapy, IVT), or into the thecal sac of the spinal cord (intrathecal therapy, ITT) are available options for certain antimicrobials with an established safety profile for that route. These, in fact, are sometimes preferentially recommended when the drug of choice is not expected to efficiently enter the BBB [24,27,28,29]. Such administration routes have been utilized for established agents, with well-defined pharmacokinetic parameters [24,30,31], but with the increasing prevalence of multidrug-resistant (MDRO) and extensively drug-resistant (XDRO) organisms, newer agents are being administered in combination therapies using innovative approaches of IV and/or IVT/ITT routes to achieve favorable outcomes in critically ill patients [28,32,33,34,35,36,37].

From a practical standpoint, treatment of CNS infections clinically relies on early identification and prompt institution of empiric antimicrobial therapy targeted against the most likely organisms, with antimicrobial resistance accounted for until sensitivities are identified. From a PK/PD perspective, essential tools in this fight against CNS infections include accurate bacterial MIC determination by the microbiology laboratory, the availability of therapeutic drug monitoring, and the ability to administer antimicrobials in alternative routes to IV, namely IVT or ITT, together with considerations for use of altered dosing strategies for optimal efficacy and favorable clinical outcomes.

In this paper, in addition to describing the available PK/PD information specific to each of the antimicrobials unitized in CNS infections, we present a synopsis of the clinical perspectives with respect to those data. As a conclusion to each section, a clinically relevant table is included for referencing the most utilized dosing regimens together with a synopsis of the agent’s PK/PD data.

## 2. Beta-Lactam Antibiotics

Beta-lactam antibiotics, defined by the presence of a beta-lactam ring, include penicillins, cephalosporins, the monobactam aztreonam, and carbapenems. The spectrum of activity varies considerably by antibiotic within each of the beta-lactam classes, and typical coverage of meningitis-associated organisms is listed in Table 1. Beta-lactams are hydrophilic molecules and are highly ionized at a physiologic pH of 7.4 systemically and at 7.3, a typical pH within the CSF [19,38,39,40]. Together, these two chemical properties limit their penetration into the CSF through the intact blood–brain barrier [19,39,40,41,42]. All beta-lactam antibiotics have a four-membered beta-lactam ring as the carbon backbone [43]. Penicillins and carbapenems contain asymmetric centers at C-5 and C-6, while the cephalosporins’ asymmetric centers are at C-6 and C-7 [43]. The various substitutions at different locations in the molecular structures lead to the differences in the spectrum of activity across beta-lactams in general and between the drugs in each subclass. By manipulation of the side chains (R), the different individual antibiotics are created under each of the parent molecule. Penicillins were the first to be discovered in 1929, followed by cephalosporins, fully characterized in 1961. The discovery of monobactam and carbapenems followed. They all share the four-membered beta-lactam ring (highlighted in blue in Figure 1). This ring provides the intrinsic antibacterial properties to all beta-lactams, due to its binding and inhibition of bacterial penicillin-binding proteins. This leads to abortive synthesis of the cell wall, hence bacterial lysis and death. The molecular structure of the monobactam aztreonam has the beta-lactam moiety as the only ring. In penicillins and carbapenems, the beta-lactam ring is fused to another 5-membered ring, whereas in cephalosporins, the other ring is 6-membered. Carbapenems differ from the other beta-lactams in that they do not possess a sulfone moiety [44,45].

Penicillin G crosses through an active transport system within cerebral capillaries leading to more rapid entry into the CSF and a shorter duration of effective concentrations [11,19,46]. Other beta-lactams cross into the CSF through paracellular pathways exhibiting peak CSF concentrations that are delayed relative to serum levels [19]. Penetration improves with inflammation and in purulent meningitis as the pH decreases from the typical CSF pH of 7.3 to a pH of 7.0 allowing for beta-lactams to cross more readily into the CSF [39,47,48]. Targeting early therapeutic concentrations with large systemic doses is necessary to ensure adequate concentrations at the site of infection, expecting that the rate of penetration would decrease as inflammation improves [18,49]. Additionally, the degree of inflammation affects free drug concentrations in the CSF due to the higher protein content during infection and may reduce bactericidal effects [49,50]. Despite that, highly protein bound drugs, such as ceftriaxone, were found to have a significantly lower rate of protein binding in the CSF compared to serum (CSF 18.8% ± 6.21% vs. serum 32.3% to 95%); this effect may be due to a saturation of binding sites when higher doses are utilized [51,52].

Within the penicillins, penetration is as low as 1% with intact (non-inflamed) BBB and may reach above 30% in the presence of inflammation [18,53]. Penicillin G has approximately two-thirds of its CSF elimination occurring via an efflux pump [19]. Inflammation inhibits the penicillin efflux pump leading to higher concentrations initially with a decrease as the inflammation improves [19]. Other beta-lactam antibiotics show a lower affinity for these active transport systems and have CSF elimination that is minimally affected by changes in inflammation [19,39]. Beta-lactams have demonstrated longer elimination times from CSF compared to serum, which may provide less fluctuation in drug concentrations compared to other sites of infection and may also provide sub-MIC effects for some pathogens [14,19].

Ampicillin has long been part of empiric meningitis treatment, particularly for niche coverage against *Listeria monocytogenes* in adults 50 years and older [54]. In an early study of ampicillin and amoxicillin, Clumeck et al. found that ampicillin was able to reach therapeutic CSF concentrations in healthy volunteers without the presence of inflammation and reached concentrations higher than amoxicillin. The authors pointed to the significant advantage of penetration through an intact BBB beyond 48 h of therapy as clinical improvement occurs and the BBB normalizes [55]. Burgess et al. performed a more recent PK/PD simulation of ampicillin and multiple other antibiotics; they found that while penicillin G demonstrated better potency based on MIC90, the pharmacokinetics of ampicillin demonstrated a longer half-life and higher unbound serum concentrations leading to a preference for ampicillin over penicillin G [56]. The addition of sulbactam further improves the bactericidal effect of ampicillin against beta-lactamase producing strains of *H. influenzae* [57]. Sulbactam has higher penetration into the CSF in the presence of inflammation, which has been found to decrease with a normal BBB or viral meningitis [57,58].

Piperacillin/tazobactam was evaluated by Ullah et al. in a simulation model of stroke patients [59]. The authors found a delay in time to reach CSF upon initial dosing. They also found that even with more aggressive dosing regimens, pathogens with an MIC above 0.5 microg/mL (*L. monocytogenes*, *S. aureus*, *E. coli*, *S. epidermidis*, *P. aeruginosa*) were unlikely to be eradicated in the CNS using piperacillin/tazobactam [59]. Evaluation of the kinetics of piperacillin–tazobactam by Nau et al. in the CSF of patients with hydrocephalus demonstrated a tazobactam concentration in the CSF lower than the desired 4 mg/liter that is required to reduce the piperacillin MIC against some Gram-negative pathogens [60]. Hence, to achieve a higher concentration of tazobactam in the CSF, the authors concluded that currently utilized doses of tazobactam in the commercially available combination formulation may not be effective in treating CNS infection. Hence, this antimicrobial combination would be inappropriate for the prophylaxis and treatment of most CNS infections, including *Pseudomonas* species.

First and second generation cephalosporins also generally do not achieve adequate concentrations in the CSF for effective utilization clinically [14]. Cefazolin is commonly used for external ventricular drain prophylaxis, and when concentrations were studied in this patient population, CSF concentrations were expected to be adequate for organisms with lower MICs but likely inadequate with standard dosing regimens (e.g., 2 g IV every 8 h) for organisms with higher MICs [61]. In a retrospective comparison between intravenous cefazolin versus cloxacillin for staphylococcal meningitis, high dose cefazolin (6 to 12 g) administered via continuous infusion was able to achieve targeted concentrations; the authors recommended using this dosing strategy paired with therapeutic drug monitoring to ensure target attainment when clinically indicated [62].

Third generation cephalosporins achieve better penetration and often maintain prolonged concentrations above the mean bactericidal concentration (MBC), which may be particularly desirable in managing infections secondary to *Enterobacteriaceae* [14,39]. Ceftriaxone and cefotaxime have both been studied in bacterial meningitis, as they possess identical antimicrobial coverage [63]. While both agents demonstrated similar CSF concentrations, ceftriaxone had a higher level of AUC_CSF_:AUC_serum_ ratio. The authors also noted the importance of administering cefotaxime not longer than every 8 h apart to maintain its therapeutic effects due to its rapid elimination. Ceftriaxone is highly protein bound in the serum (83–96%), likely leading to the delayed entry in the CSF, but it provides benefit through a long half-life in both the serum and CSF [19,52]. In an early pediatric comparison study between ceftriaxone and ampicillin with chloramphenicol, del Rio et al. found greater bactericidal activity in the CSF with ceftriaxone [64]. Experimental models have also shown that the BBB penetration of ceftriaxone was unaffected by the use of steroids [14]. These characteristics have likely led to ceftriaxone being the most utilized beta-lactam in the treatment of meningitis. Cefotaxime, while used less commonly for the treatment of meningitis in adults, remains a viable monotherapy option for susceptible organisms (e.g., *S. pneumoniae*) [65]. This is especially true in the pediatric population, where it has been the preferred third generation cephalosporin due to a relatively more favorable safety profile [66,67].

Ceftazidime and cefepime have a role in therapy for empiric or definitive treatment of hospital-acquired infections. Kassel et al. found that utilizing cefepime every 8 h had a higher target achievement of fT > MIC ≥ 60% for an MIC of 8 mcg/mL (70% vs. 20% for every 12 h regimen; *p* = 0.02) [68]. Nau et al. studied ceftazidime pharmacokinetics in patients with external ventriculostomies and considered its use would be suboptimal, especially in pathogens with higher MIC values (e.g., *Pseudomonas*) [69]; if used, the authors recommended combination therapy with an aminoglycoside. Standard dosing regimens of cefepime and ceftazidime are likely unable to achieve adequate CSF concentrations with increasing MICs of hospital-acquired pathogens and warrant more aggressive regimens or utilization of continuous infusion [24]. Ceftaroline is one of the newer cephalosporins approved by the FDA and has a desirable extension of the beta-lactam spectrum of activity to include methicillin-resistant *Staphylococcus aureus* (MRSA). Ceftaroline has had limited evaluation for CNS infections but has been used off-label for this indication [70]. Similar to the other beta-lactams, it has low CNS penetration. In a Monte Carlo simulation, Helfer et al. found that ceftaroline had 90% predicted target attainment for fT > MIC of 28.8% with an MIC of 1 mg/L; this improved to 99.8% and 97.2% in the presence of inflammation when ceftaroline 600 mg was administered every 8 h and every 12 h, respectively [70].

Aztreonam differs from traditional beta-lactam antibiotics in that it has only one beta-lactam ring, which reduces the allergic cross-reactivity rate to less than 1% of patients with a beta-lactam allergy [71,72]. Patriarca et al. found no cross-reactivity in 45 patients with a history of one or more beta-lactam allergies, but the potential for cross-reactivity with ceftazidime may be higher as they share identical side chains [73,74]. Structural differences from other beta-lactams also affect the spectrum of antimicrobial coverage. The aminothiazolyl oxime side chain provides activity against Gram-negative bacilli, while a carboxyl side chain allows for enhanced activity against *Pseudomonas aeruginosa*, even in multidrug-resistant strains [75,76,77]. An a-methyl group at position four allows for stability in the presence of beta-lactamases [75,76,77]. Aztreonam preferentially binds to penicillin-binding protein 3 (PBP-3) of the Gram-negative bacterial cell wall. Since most Gram-positive and anaerobic organisms lack PBP-3, aztreonam has poor affinity for Gram-positive and anaerobic organisms [78]. This narrow spectrum of activity has led to limitation of the clinical utility of aztreonam to primarily being an alternative agent when aminoglycosides are indicated but cannot be utilized [79]. Aztreonam is currently the only clinically available member of the monobactam class and has been identified as one of the older antibiotics with potential to treat multidrug-resistant organisms [80]. Aztreonam is bactericidal against common causes of Gram-negative meningitis, including *H. influenzae*, *P. aeruginosa*, and *E. coli*, while also having good activity against *Neisseria meningitidis* [81]. The mean CSF concentrations of aztreonam in patients with non-inflamed meninges were 5–10 times the concentrations necessary to inhibit most *Enterobacteriaceae*. The concentration in CSF in patients with inflamed meninges was four times higher than those with non-inflamed meninges [82].

Carbapenems have a better rate of CSF penetration compared with other beta-lactams. Meropenem penetrates the CSF and achieves therapeutic levels in patients with inflamed meninges [47]. Imipenem, while able to achieve therapeutic levels within the CSF, is not typically used due to the increased risk for CNS adverse effects, such as epileptic seizures [83]. In spite of these adverse effects, imipenem/cilastatin has been used to treat pneumococcal meningitis when failure of third generation cephalosporins occurs [84]. Fewer data are available for newer carbapenems, such as ertapenem and doripenem, regarding penetration into the CNS [11]. A preliminary study shows that doripenem penetrates the BBB to a small extent; however, more studies are needed before any recommendations can be made to utilize this antimicrobial agent in the treatment of CNS infections [85]. Once reaching the CSF, carbapenems do not diffuse as easily through the cell wall as other beta-lactam antibiotics; they instead enter through outer membrane proteins called porins and are then able to bind to PBPs [86,87].

Newer beta-lactam/beta-lactamase inhibitor (BLBLI) combinations have been evaluated as potential treatments for Gram-negative infections. Sime et al. examined ceftolozane–tazobactam penetration after a single dose and proposed that the concentrations would be inadequate with maximum dosing to treat Gram-negative meningitis as monotherapy [88]. However, several cases have been published using ceftolozane–tazobactam as part of a successful salvage therapy regimen for multidrug-resistant (MDR) Gram-negative infections [89,90,91,92]. Ceftazidime–avibactam has also been used successfully for treatment of MDR Gram-negative infections including carbapenemase-producing *Enterobacteriaceae* [93,94,95,96]. While further studies are needed, the use of the novel BLBLI combinations remains an option for MDR CNS infections.

Table 1 summarizes the PK/PD data of beta-lactam antibiotics used in the management of CNS infections. 

**Table 1 antibiotics-11-01843-t001:** Beta-lactam antimicrobial drugs and their PK/PD data.

Drug	CSF/Serum ^a^ (%)	Serum Protein Binding	Primary Route of Elimination	Serum Elimination Half-Life	Serum C_max_	Systemic Dosing	Spectrum of Activity							
							*S. pneumoniae*	*S. agalactiae*	*S. aureus* MS/MR	*H. influenzae*	*E. coli*	*P. aeruginosa*	*N. meningitidis*	*L. monocytogenes*
Beta-lactams														
Penicillin G	5–10	~60%	Renal (58–85% unchanged)	31 to 50 min	400 mg/L	4 million units IV every 4 h	+	+	−/−	−	−	−	+	+
Ampicillin	13–14	15 to 18%	Renal (~90% unchanged)	1 to 1.8 h	109 to 150 mg/L	2 g IV every 4 h	+	+	−/−	−	−	−	+	+
Nafcillin	<0.2–20	~90% (primarily albumin)	Feces, urine (30% unchanged)	33 to 61 min	~30 mg/L	2 g IV every 4 h	+	+	+/−	−	−	−	−	−
Oxacillin	1.0–2.8	~94% (primarily albumin)	Urine and bile (unchanged)	20 to 30 min	43 mg/L	2 g IV every 4 h	+	+	+/−	−	−	−	−	−
Piperacillin	1.8–32	~16%	Urine	~1 h	108.2 ± 31.7 mg/L ^c^	NR								
Cefazolin	0–4	80%	Urine (70–80% unchanged)	1.8 h	94 ± 30.33 mg/L	2 g IV every 8 h (Novak 2021 CI: 6–12 g per day over 24 h	+	+	+/−	−	+	−	−	-
Cefoxitin ^d^	0.8–35	65 to 79%	Urine (85% unchanged)	41 to 59 min	110 mg/L	NR								
Cefuroxime ^e^	11.6–13.7	33 to 50%	Urine (66–100% unchanged)	~1 to 2 h	100 mg/L	NR								
Cefotaxime	3–48	31 to 50%	Urine (60% unchanged)	1 to 1.5 h	214.4 mg/L	8–12 g/day divided every 4–6 h	+ ^b^	+	+/−	+	+	v	+	−
Ceftriaxone	0.6–94	85 to 95%	Urine (33–67% unchanged)	~5 to 9 h	280 ± 39 mg/L	2 g IV every 12 h	+ ^b^	+	+/−	+	+	−	+	−
Ceftazidime	2.7–15	<10%	Urine (80-90% unchanged)	1 to 2 h	61.9 to 79 mg/L	2 g IV every 8 h	−	+	−/−	+	+	+	+	−
Cefepime	10	~20%	Urine (85% unchanged)	2 h	129 ± 27.1 mg/L	2 g IV every 8 h CI: 0.5 g over 30 min followed by 4 g over 24 h	+	+	+/−	+	+	+	+	−
Ceftaroline	0.5–4.3	~20%	Urine (88% unchanged)	1.6 to 2.7 h	22.3 ± 5.9 to 22.6 ± 2 mg/L	600 mg every 8-12 h	+	+	+/+	+	+	−	+	
Ceftolozane	20–40	16 to 21%	Urine (>95% unchanged)	~3 to 4 h	73.9 ± 25.4 mg/L	Variable and limited data; 3 g ceftolozane–tazobactam over 1 h every 8 h Potential off-label doses up to 4.5 g and administration as prolonged infusion over 3 h or CI	+	+	−/−	+	+	+		
Aztreonam	1–37	~77%	Urine (60%–70% unchanged) Feces (~12%)	2.1 h	204 mg/L	6–8 g/day divided every 6–8 h	−	-	−/−	+	+	+	+	−
Imipenem	1–45	~20%	Urine (~70% unchanged)	~60 min	44.2 ± 13.26 mg/L	NR due to neurotoxic effects								
Meropenem	10.7–21	~2%	Urine (~70% unchanged, ~28% inactive metabolite) Feces (2%)	1 h	~49 mg/L (39 to 58 mg/L)	2 g IV every 8 h	+		+/−	+	+	+	+	+
Beta-lactamase Inhibitors														
Avibactam	38	5.7 to 8.2%	Urine (97% unchanged)	2.7 h	12 to 15.5 mg/L									
Clavulanate	6–17	~25%	Urine (25–40% unchanged)	1 h	2.4 ± 0.83 mg/L									
Sulbactam	13.5 (Wang 2015)	38%	Urine (75–80% unchanged)	1 to 1.3 h	48 to 88 mg/L									
Tazobactam	3–74	30%	Urine (>80% unchanged)	~2 to 3 h	21.7 ± 7.8 mg/L									
Vaborbactam		~33%	Urine (75–95% unchanged)	1.68 h	55.6 ± 11 mg/L									

CSF, cerebral spinal fluid; t1/2, half-life; h, hours; g, grams; IV, intravenous; AUC, area under the curve; NR, not recommended; CI, continuous infusion; MS, methicillin-sensitive; MR, methicillin-resistant; ^a^. CSF/serum concentrations will vary depending on inflamed conditions (e.g., meningitis); ^b^. ceftriaxone or cefotaxime plus vancomycin for *Streptococcus pneumoniae;*
^c^. piperacillin and tazobactam Cmax following 4 h infusion; ^d^. cefoxitin package insert. Mylan Institutional LLC, 2017; ^e^. claforan. Package insert. Sanofi-Aventis U.S. LLC, 2015; References [14,19,24,54,61,63,97,98,99]

The bactericidal activity of beta-lactam antibiotics is time-dependent; it is determined by the amount of time that the free drug concentration remains above the minimum inhibitory concentration (fT > MIC) for the organism [16,40]. Ideally, efficacy is improved when fT > MIC for at least 50% of the time between doses and recommended at 100% in immunocompromised individuals. Additionally, when the antibiotic concentration is 4–5 times above the MIC of the organism being targeted, efficacy is further improved. This proportion of time above the MIC is best attainable for cephalosporins and aztreonam as compared to penicillins, and it is better achieved for penicillins than it is for carbapenems [100,101].

In contrast to bactericidal activity in serum, proposed slower bacterial growth in the CSF may reduce beta-lactam activity as they rely on cell wall synthesis and rapid bacterial multiplication for maximum effect [14,39,49,50]. Moreover, an impaired immune response requires minimum bactericidal concentrations to be reached rather than only inhibitory concentration [49]. Lutsar et al. found the best linear correlation occurred between time above the mean bactericidal concentration (T > MBC) and bacterial killing rate in an experimental rabbit meningitis model using ceftriaxone [19,102]. The authors proposed that high concentrations above MIC or MBC may not be necessary, and that beta-lactams maintain time-dependent bactericidal activity in the CSF similar to the effects in the serum [19]. In this experimental model, they found that dividing the same ceftriaxone dose into two doses per day provided continued bactericidal activity throughout the time period compared with the total dose at once, which led to a cessation of the killing effect after 12 h.

The dosing of beta-lactam antibiotics is empirically selected based on the expected PK profile of the individual drug and expected limited CSF penetration. Generally, beta-lactam antibiotics are considered to have a wide therapeutic index for safety to allow for more aggressive dosing regimens in CNS infections without the need to limit or decrease doses to avoid toxicity [17,40,103]. With the increasing prevalence of drug-resistant organisms and elevated MIC targets, dose escalation has been common, bringing into question the threshold for toxicity with supratherapeutic exposure versus clinical failure with subtherapeutic concentrations. Therapeutic drug monitoring (TDM) studies for beta-lactams have been published since 2009, but unlike other antibiotics (e.g., aminoglycosides, vancomycin), TDM for beta-lactams is still not widely available for clinical use. This can increase the risk of either clinical failure or toxicity if the empiric dosing regimen is not correctly selected [20,104]. A study by Udy et al. evaluated unbound beta-lactam concentrations and found creatinine clearance (CrCl) was a statistically significant contributor to whether a therapeutic concentration was obtained. The authors demonstrated trough levels less than MIC in 82% and less than four times the MIC in 72% of patients with CrCl ≥ 130 mL/min/1.73 m^2^ (*p* < 0.001; *p* < 0.001, respectively) [20,105]. Roberts et al. similarly found that 74.2% of initial doses provided inadequate steady state concentrations for maximum effects; patients with meningitis in this study required a dose increase in 47% of the cases [104]. TDM remains an important opportunity for future research to correlate target CSF concentrations with clinical effectiveness against pathogens including resistant ones, tailoring that to patient specific factors such as obesity, immunocompromised states, extracellular fluid deviations, and augmented renal clearance, with the intent of the optimization of PK/PD targets [40,104,106,107].

Recently, discussion has increased on the minimum and maximum acceptable doses to ascertain both efficacy and safety with this class of antibiotics [108]. Neurotoxicity from beta-lactams is likely related to the beta-lactam ring and its binding affinity at GABA receptor sites leading to inhibition on GABA neurotransmission [108,109]. Neurotoxicity risk increases with higher doses as used in treating meningitis [49]. Other risk factors include renal or hepatic insufficiency, hypoalbuminemia (i.e., increased free drug availability), advanced age, and other CNS disorders or predisposing conditions that increase BBB permeability (e.g., stroke) [40,109]. The risk of neurotoxicity appears to be highest with penicillin G, cefazolin, cefepime, ceftazidime, and imipenem [40]. Historically beta-lactams were trialed for IVT administration in an attempt to circumvent poor CNS penetration. Administration of beta-lactams intraventricularly was linked to harmful CNS effects including seizures and in the most severe cases, death [19]. IVT administration of beta-lactams is not recommended.

To maximize the time above MIC or MBC, the continuous infusion administration of beta-lactams has been evaluated. In an experimental rabbit model, no difference was seen in intermittent infusion versus continuous infusions of penicillin G; the use of an every 4 h intermittent dosing schedule was able to adequately maintain concentrations above the MBC [14,39]. One theory proposed that brief exposure to subinhibitory concentrations may allow for bacterial regrowth and optimize the beta-lactam mechanism of action [39,48]. However, with an increase in bacterial resistance, the use of continuous infusions may still be beneficial in the management of MDRO or patients with CNS infections lacking inflammation (e.g., ventriculitis) to optimize the fT > MIC. Meropenem has been studied to determine if continuous infusions can be utilized to treat CNS infections, and it has been found that infusion rates of 125 mg/h and 250 mg/h achieved sufficient concentrations greater than the MIC for susceptible organisms and intermediately resistant organisms [110]. However, the short room temperature stability of ~4 to 6 h does make continuous infusions more difficult to manage [111]. Huang et al. found that use of a bolus followed by continuous infusion with cefepime was able to achieve higher AUC_CSF_:AUC_plasma_ ratios compared with intermittent infusion (18.4% vs. 9.7%) [112]. They also found that the concentrations remained above MIC for greater than 75% of the time with MICs of 8 mg/mL compared with 0% of the time with intermittent dosing [112]. Grégoire et al. recently published a dose optimization nomogram to improve ceftriaxone dosing based on renal function to avoid underdosing (Figure 2) [113]. For example, within the nomogram, an 80 kg patient with an estimated glomerular filtration rate (eGFR) of 50 mL/min/1.73 m^2^ may achieve adequate concentrations with 2 g ceftriaxone twice daily, but if the same 80 kg patient had an eGFR > 110 mL/min/1.73 m^2^, the same dose would be expected to be subtherapeutic. Future studies should pursue a similar focus on beta-lactam dose optimization and TDM with concentration targets correlated to clinical outcomes.

Carbapenems are unique in comparison to other members of the beta-lactam family in that they exhibit a post-antibiotic effect (PAE). Meropenem in particular has been shown to have a PAE up to 2.5 h when *Pseudomonas aeruginosa, Staphylococcus aureus*, and *Enterobacteriaceae* were exposed to drug concentrations that were four times the MIC for 1.5 h [21]. Meropenem’s PAE has been shown to be extended when used in combination with gentamicin [114].

### Clinical Perspectives in Consideration of Beta-Lactam PK/PD Data

Some beta-lactam antibiotics are excellent options for CNS infections as empiric therapy, but also as definitive therapy when the infectious agent and its sensitivities are identified. They are utilized intravenously, with no role for ITT or IVT administration, as this has been shown to be harmful. Of all beta-lactams, ceftriaxone is the primary and most utilized beta-lactam in CNS infections, specifically in bacterial meningitis and brain abscess. Generally, the dose for treatment of active disease is standard and requires no adjustment in renal and hepatic failure but may require an increase in dosing with increased eGFR above eGFR > 110 mL/min/1.73 m^2^. Penicillin G is the primary choice for neurosyphilis and for susceptible *Neisseria meningitidis* strains, as well as an alternative agent to ceftriaxone for bacterial meningitis with susceptible bacteria. Ceftazidime and cefepime are agents of choice in catheter or shunt-related bacterial meningitis in combination with non-beta-lactam agents for synergy. Piperacillin–tazobactam has no standard role in the treatment of CNS infections. Aztreonam, a beta-lactam antimicrobial with only one ring, is distinguished in clinical practice as primarily useful in the treatment of Gram-negative infections, including *Pseudomonas*, without activity against Gram-positive organisms. With its excellent tissue penetration, including the meninges across the BBB in the inflamed and non-inflamed statuses, aztreonam has a specific clinical niche in managing CNS infections such as meningitis when *H. influenzae*, *E. coli*, *Neisseria meningitidis*, or *P. aeruginosa* [115] are suspected or confirmed, where it may be utilized as monotherapy. However, it can also be added to vancomycin if Gram-negative organisms are suspected in ventriculitis, where they may occur in 5–8% of CNS surgical site infections. Aztreonam also replaces aminoglycosides when these agents are indicated but cannot be utilized [116]. Meropenem has a primary role in cases of bacterial resistance or contraindication to other beta-lactams. It is preferred over imipenem due to imipenem’s propensity to decrease the seizure threshold in patients with risk factors for seizure or with convulsive disorders. Inflammation of the meninges facilitates entry of beta-lactams from serum to the CSF, but that does not necessarily translate into more effective antimicrobial activity because of the factors discussed above.

## 3. Vancomycin

Vancomycin is a large, hydrophilic glycopeptide that is one of the most commonly used antibiotics, via several administration routes, IV, IVT, and ITT, for the empiric treatment of CNS infections [117,118]. The bactericidal activity of vancomycin is both concentration- and time-dependent, related to the ratio of the area under the concentration time curve (AUC) for the free (non-protein bound) fraction of the drug to the MIC [24]. Although both pharmacokinetic parameters fT > MIC and Cmax are important in determining the therapeutic efficacy of vancomycin, the AUC/MIC ratio is the major determinant of its therapeutic efficacy [16,119]. Vancomycin’s therapeutic efficacy may indeed be more concentration-dependent than it is time-dependent [19,119,120]. There does not appear to be any difference in patient outcomes between vancomycin administered by continuous infusion or by intermittent administration, thus supporting the notion that vancomycin is more concentration-dependent [121].

Reported protein binding of vancomycin ranges between 30 and 60%, which could contribute to inadequate drug disposition into the CNS [48,122]. In the absence of meningeal inflammation, the penetration into the CSF of vancomycin is hampered by its high molecular weight and hydrophilicity [48,53,123,124]. With inflammation, the tight junctions of the blood–brain barrier cells are damaged, which facilitates entry into the CSF. While penetration in inflamed meninges is reportedly as high as 81%, and reports on penetration into the CSF in normal or mildly infected meninges are conflicting, found to vary between 0 and 36% [10]. It is estimated that with meningitis, vancomycin achieves a level in CSF up to 22% of that in the serum [14]. Vancomycin penetration into the CSF is slower than the clearance from the CSF. This is seen in studies in patients with ventriculitis as well as in subjects with non-inflamed meninges [30,123,125]. Wang and colleagues investigated whether the CSF concentration of intravenously administered vancomycin reached therapeutic levels following neurosurgery, where disruption of the BBB is expected. Twenty-four hours after surgery, vancomycin administration achieved a peak concentration of 4.4 mg/L in one patient. Another patient, given the dose 72 h post-operatively, had a peak of 11.9 mg/L. This demonstrates that the disruption of the blood–brain barrier (BBB) after neurosurgical procedures may be prolonged and increase the penetration of intravenously administered vancomycin with the subsequent increase in CSF concentration [48,126].

Blassmann and colleagues reported that vancomycin achieves adequate CSF concentrations after IV administration, with dose increases being required in the setting of augmented renal clearance [20,123]. Vancomycin concentrations in CSF are at least partially dependent on the level of meningeal inflammation with relatively low CNS penetration overall and substantially variable CNS concentrations following systemic dosing alone [10,27,123]. Ricard and colleagues determined that concomitant dexamethasone (10 mg every 6 h) did not affect vancomycin therapy (continuous infusion of 60 mg/kg/day after a 15 mg/kg loading dose) because acceptable levels of vancomycin were obtained in the CSF (mean value 7.9 mg/L) [12]. High doses of vancomycin are required to achieve optimum serum and CSF vancomycin concentrations in patients with ventricular drainage [53,117], a procedure indicated in certain neurosurgical conditions (such as hematoma, elevated intracranial pressure, acute hydrocephalus, and sometimes in meningitis).

Vancomycin can also be used synergistically with other antibiotics, such as with ceftriaxone for pneumococcal meningitis [14]. Rifampin can be considered in addition to vancomycin for staphylococcal CNS infections if the organism is susceptible and prosthetic material is also in place and for *Streptococcus pneumoniae* CNS infections if the MIC to ceftriaxone is >2 ug/mL [31].

There is less clarity about the relevant drug exposure at the target site of infection and the regimens required to achieve these targets [123]. Blassman and colleagues reported poor penetration of vancomycin into CSF in patients with proven or suspected ventriculitis with a median CSF/serum ratio of 3% with high interpatient variability, leading to the belief that therapeutic drug monitoring of both serum and CSF may be needed to optimize therapy. The ASHP Therapeutic Position Statement on therapeutic monitoring of vancomycin in adult patients recommends the optimal vancomycin serum trough concentration for CNS infections is 15–20 mg/L in order to improve CSF penetration, increase the likelihood of reaching optimal target serum concentrations, and improve clinical outcomes [121]. Guidelines recommend dosing vancomycin at 30–60 mg/kg/day for meningitis and ventriculitis to ensure sufficient CSF concentrations [31,121,123]. Albanèse and colleagues reported successful treatment of bacterial meningitis utilizing continuous vancomycin infusion at a mean dose of 62 mg/kg/day to obtain serum concentrations of 25–30 mg/L and CSF levels of 6–19 mg/L [125]. ITT administration of vancomycin is also an option as there have been very few side effects reported and no contraindications for this route. ITT with vancomycin dosed at 10–20 mg every 24 h will ensure concentrations above the MIC of susceptible pathogens for the entire dosing interval [25], and the IDSA guidelines recommend doses from 5 to 20 mg/kg [31]. In a recent meta-analysis by Schneider and colleagues looking at the efficacy of vancomycin in CNS infections, no superior dosing regimen could be identified for meningitis or ventriculitis [124]. There is a need for better defined clinical outcomes, optimal pharmacokinetic/pharmacodynamics, and toxicodynamic parameters following vancomycin administration for CNS infections [122].

The IDSA guidelines recommend IVT antimicrobial therapy for patients with healthcare-associated ventriculitis and meningitis in which the infection responds poorly to systemic antimicrobial therapy alone [31]. IVT dosing is likely necessary for the treatment of ventriculitis allowing substantially higher concentrations [24]. A systematic review by Beach and colleagues demonstrated no relationship between the overall CSF levels of vancomycin and clinical/microbiological cure of ventriculitis [10]. CSF sterility and normalization of CSF parameters have been achieved sooner with the use of intraventricular therapy and intravenous therapy together as compared to intravenous therapy alone [31]. Nau and colleagues report utilizing IVT vancomycin at a dose of 5–20 mg/kg every 24 h may result in temporary hearing loss [11].

### Clinical Perspective in Consideration of Vancomycin PK/PD Data

Vancomycin is a glycopeptide that is very commonly used in CNS infections intravenously, but also intrathecally and intraventricularly for patients with resistant organisms. It is safe and effective by all these routes, with the pharmacodynamic activity via time-dependent and concentration-dependent bacterial killing. The drug is administered two or three times per day, but also continuous infusions are utilized to enhance the AUC maintenance above the MIC for the duration of the time of utilization. The primary spectrum of activity is against Gram-positive organisms, and hence it is used either as monotherapy for documented MRSA infections, such as with hospital associated meningitis, ventriculitis, or other shunt-related CNS infections, but also as combination therapy with beta-lactams for empiric therapy against community-acquired CNS infections until antimicrobial resistance is ruled out. Its penetration into the BBB is enhanced by meningeal inflammation or ventriculitis, as well as after neurosurgical procedures. Entry into the CSF is much less efficient when there is no CNS inflammation. Its activity, penetration, and levels in the CSF are not affected by concomitant use of dexamethasone. Additionally, multiple studies have failed to demonstrate a direct relationship between its degree of microbial killing vis-à-vis CNS levels. TDM is a tool utilized in clinical practice to monitor its levels and ensure a therapeutic serum concentration. Table 2 summarizes the PK/PD data of vancomycin.

## 4. Aminoglycosides

Aminoglycosides are active in vitro against Gram-negative organisms including most *Enterobacteriaceae* and *Pseudomonas aeruginosa* [22]. They exhibit concentration-dependent killing; the higher the drug concentration relative to pathogen minimum inhibitory concentration (MIC), the greater the rate and extent of antimicrobial activity. Aminoglycosides also exhibit PAE, which leads to persistent suppression of bacterial growth long after administration is complete [16,19]. They have limited access to the CNS due to the BBB due to their hydrophilicity and poor penetration even in the presence of meningeal inflammation [18,22]. Penetration of aminoglycosides in the presence of significant meningeal inflammation remains poor because brain capillaries lack the basement membrane pores of systemic capillaries rendering them impermeable to the aminoglycosides’ large hydrophilic molecules [14,16,128]. Systemic administration of aminoglycosides when used as monotherapy does not achieve effective blood levels [14]. High dose IV therapy is limited by their narrow therapeutic range due to nephrotoxicity and ototoxicity and achieves too low CSF concentrations (0.1–0.45 mg/liter) to be clinically relevant [17,24]. The reported risks of concurrent vancomycin and aminoglycoside administration in humans provides conflicting information on whether there is no effect or enhanced nephrotoxicity. Rybak and colleagues found that patients who received both agents concurrently were almost 7-fold more likely to develop nephrotoxicity [129]. According to most of the available published data, it appears there is a 3- to 4-fold increase in nephrotoxicity when these agents are used in combination. The incidence of ototoxicity may increase when aminoglycoside therapy is used in addition to vancomycin. In these instances, monitoring of these agents is important for the prevention of these toxicities from occurring [121].

Use of aminoglycosides may require direct instillation by IVT or ITT administration into the cerebrospinal fluid (CSF) to achieve therapeutic levels at the infection site while limiting systemic toxicity [14,17,48]. IVT or ITT administration may be considered when IV administration alone fails to achieve a clinical or laboratory response of bacterial meningitis caused by susceptible organisms [22,122]. Case reports describe high CSF concentrations post IVT doses (>100 mg/L, 1.59 mg/L) in comparison to IV administration, which did not achieve CSF concentrations above 0.5 mg/L [24]. In a case series of 14 patients with bacterial meningitis, survival, meningitis cure and CSF sterilization rates of 31, 64, and 86% were demonstrated with IVT therapy used in combination with IV aminoglycosides [130]. Notably, defined pharmacokinetic/pharmacodynamic and toxicodynamic targets for aminoglycosides in the CSF are absent from the published literature, complicated by the lack of applicability for using systemic aminoglycoside levels as surrogate markers. Similarly, there is limited guidance available for aminoglycoside drug monitoring in the CSF [122]. Optimal dosing regimens for IVT therapy remain unclear with each drug having a range of reported doses (amikacin 5–50 mg daily, tobramycin 5–20 mg daily, gentamicin 4–20 mg daily) and a wide range of duration of therapy (3–40 days). A lack of prospective clinical trial data on the IVT administration of aminoglycoside use and the risk of adverse effects, such as temporary hearing loss, seizures, aseptic meningitis, and eosinophilic CSF pleocytosis, lead to this route being reserved for seriously ill patients for whom systemic antimicrobials have failed to eradicate the infecting organism or those with recurrent infection [22]. IVT administration has demonstrated increased mortality in some neonatal studies [48].

There have been reports of ITT administration of aminoglycosides that led to CSF sterilization and lower mortality. There were no significant side effects reported for gentamicin or tobramycin, but there were reports of hearing loss and tonic-clonic seizures post-amikacin ITT administration. Optimal dosing regimens for ITT therapy remain unclear with each drug having a range of reported doses (amikacin 4–50 mg daily, tobramycin 5–20 mg daily, gentamicin 1–10 mg daily) and a wide range of durations of therapy (3–180 days) [118].

### Clinical Perspective in Consideration of Aminoglycoside PK/PD Data

Aminoglycosides penetrate the CSF poorly, even with inflammation of the meninges. Together with their narrow therapeutic range and high degree of toxicity, they are of limited utility upon systemic administration in the management of CNS bacterial infections. IVT and ITT administrations are likewise of limited clinical utility, have no standardized dosing regimens, and can lead to direct CNS toxicity. Hence, aminoglycosides could be utilized as a last therapeutic frontier in patients with no other alternatives [17]. The availability of more efficacious and better tolerated antimicrobials as alternatives has rendered them less attractive for use in the context of CNS infections, for which they are rarely used in clinical practice for their management.

## 5. Linezolid

Linezolid is an oxazolidinone antimicrobial known for its activity against multidrug-resistant Gram-positive organisms, including MRSA and VRE [131]. The consensus on which agent is optimal for treating VRE *faecium* CNS infections remains to be determined [132]. Linezolid is a bacterial protein synthesis inhibitor with bacteriostatic activity against *Enterococcus* species, which has raised concerns regarding its clinical benefit, particularly when used in patients who are immunocompromised or have deep-seated infections [27]. In a systematic review comparing clinical outcomes between bacteriostatic and bactericidal agents, Wald-Dickler and colleagues concluded that in contrast to other static agents that achieved very low blood concentrations, linezolid possesses more favorable bloodstream pharmacokinetics due to having superior or no relevant differences in clinical outcomes for Gram-positive bloodstream infections when compared to bactericidal drugs such as vancomycin and teicoplanin [131]. According to two population PK studies in critically ill neurosurgical patients, linezolid reached mean CSF concentration to serum ratios of 66% and 77%, suggesting good CSF penetration [133]. However, other studies report a high interpatient variability in CSF concentrations, which threatens efficacy for organisms with higher MIC values of 2 and 4 mg/L [24], and that would suggest a higher dose may be required to achieve higher CSF levels for optimal efficacy. There has been one case report from Dietz and colleagues that described the use of linezolid 600 mg every 8 h, rather than the standard 600 mg every 12 h dosing [134]. Among 19 cases that utilized linezolid for VRE *faecium* CNS infections, 15 reported a clinical cure (78.9%) of which monotherapy with linezolid was used in 53.3% (8/15) of the cases [135,136,137,138,139,140,141,142]. There is one successful case report in the literature describing ITT linezolid for the treatment of *Enterococcus faecalis* ventriculitis [143]. The reported patient was administered linezolid intrathecally via an external ventricular drain (EVD) at 10 mg daily for a total of 15 days [143]. There have also been some reports of combination therapy with daptomycin and linezolid. The proposed mechanism is that daptomycin depolarizes the cell membrane, which may increase the access of linezolid to the target ribosome [144]. With its known myelosuppressive side effect, initially observed in clinical trial participants, and in post-marketing studies at higher rates, blood cell parameters should be closely monitored in patients on linezolid, especially with extended durations (>10 days) [27]. Based upon these data, linezolid monotherapy may be an option for the treatment of susceptible VRE *faecium* CNS infections, and it has a clinical niche as an alternative to vancomycin in the treatment of MRSA brain abscesses [145]. ITT or IVT administration cannot be recommended at this time due to the lack of supportive evidence of safety and efficacy in large numbers of patients.

### Clinical Perspective in Consideration of Linezolid PK/PD Data

Linezolid therapy may be an option for the treatment of resistant Gram-positive nosocomial ventriculitis and meningitis, specifically VRE and MRSA, and as an alternative when other agents fail clinically. It requires no hepatic or renal adjustment. It has been shown in some studies to achieve CSF levels up to 77% that of serum, although this has not been reproducible in other studies. Long-term adverse effects, particularly pancytopenias, limit its use. IVT and ITT therapies cannot be recommended at this time. Table 3 summarizes the PK/PD data relevant to linezolid.

## 6. Daptomycin

Daptomycin is a cyclic lipopeptide with concentration-dependent pharmacokinetics that exhibit rapid bactericidal activity against Gram-positive organisms, including resistant strains such as vancomycin-resistant *Enterococcus* (VRE) and methicillin-resistant *Staphylococcus aureus* (MRSA). It is bactericidal and has been successfully utilized in the treatment of VRE bacteremia and endocarditis. Daptomycin has a large molecular weight and a high degree of protein binding (>90%), which is thought to contribute to limited penetration into the CNS [27]. There are some reported treatment successes in bacterial meningitis [27,147,148]. Direct access to the CSF space via ITT or IVT daptomycin installation provides an alternative route of administration that has proven highly effective, especially when failure with IV linezolid and daptomycin has occurred [27]. The optimal dose is not established but reported treatment successes utilized IVT daptomycin 5 mg either daily or every 48 h anywhere from 2 doses to 54 days [134,149,150]. Piva and colleagues studied the penetration of daptomycin in the CSF after IV infusion at the dose of 10 mg/kg and found the CSF/serum ratio to be only 0.45%, determining that it is unlikely that IV daptomycin administration could reach effective CSF concentrations to have clinical efficacy. Effective treatment with systemic administration could be obtained with doses higher than 10 mg/kg, but there are no current studies that have evaluated these higher doses [29]. Applicability in VRE *faecium* CNS infections remains indeterminate, with a limited number of case reports finding success using different dosing strategies (synergy with other antibiotics, increased doses, etc.) [27].

### Clinical Perspective in Consideration of Daptomycin PK/PD Data

Daptomycin is a relatively newer agent, a lipopeptide rarely utilized for CNS infections. With bactericidal properties against Gram-positive cocci, it has emerged as an alternative therapy when treating MRSA or VRE CNS infections when other agents are contraindicated or have failed. IV daptomycin is unlikely to effectively cross the BBB, hence ITT or IVT administrations were evaluated and shown to be successful in some case reports. The PK/PD data relevant to daptomycin are provided in Table 4.

## 7. Metronidazole

Metronidazole is a synthetic nitroimidazole antimicrobial introduced originally to treat Trichomonas vaginalis, and one of the current mainstay drugs used to treat infections caused by anaerobic bacteria (*Bacteroides fragilis*, *Prevotella* species, *Fusobacterium necrophorum*, *Clostridium difficile*, *Gardneralla vaginalis*), protozoa, and microaerophilic bacteria. It exerts a bactericidal cytotoxic effect by introducing free radicals that damage the host DNA. This inhibits protein synthesis and induces cell death. Typically administered orally or intravenously (500 mg over 30 min every 8 h), metronidazole has concentration-dependent killing with a post-antibiotic suppressive effect [152,153]. Several studies have demonstrated rare neurotoxic effects, such as metronidazole-induced encephalopathy and autonomic neuropathy that resolve upon the discontinuation of the drug [154,155,156,157,158]. This speaks to the drug’s ability to penetrate the CNS, which is likely due to metronidazole’s lipophilicity that also renders it efficacious in the treatment of bacterial meningitis and brain abscesses [11,159,160,161,162]. Numerous studies have attempted to quantify its CNS availability. With a greater than 90% oral bioavailability and high volumes of distribution approaching 60–100% of plasma concentrations [14,163], metronidazole can effectively penetrate and treat CNS infections. CNS drug distribution has traditionally been assessed by sampling cerebrospinal fluid drug concentration via lumbar puncture or external ventricular drainage. However, a recent study by Frasca et al. utilized intracerebral microdialysis in acute brain injury patients to better quantify the distribution of metronidazole in brain parenchyma by sampling the extracellular fluid (ECF) [164]. Their results demonstrated maximal concentrations in the brain that were slightly but not significantly lower than corresponding plasma concentrations. The mean brain-to-unbound plasma ratio was equal to 102% ± 19% in brain parenchyma after the administration of 500 mg of metronidazole every 8 h. Additionally, a comparison of the concentration–time curves of the drug showed a peak concentration in the ECF comparable to that of unbound plasma concentrations [165]. This was contrary to the CSF concentration that remained essentially flat, which supports the value of new techniques in assessing CNS drug availability in addition to new evidence of the extensive CNS penetration of metronidazole.

### Clinical Perspectives in Consideration of Metronidazole PK/PD Data

Metronidazole remains a reliable CNS-penetrating antimicrobial with a unique spectrum of activity that can target susceptible anaerobic microorganisms to effectively treat brain abscesses and meningitis. Metronidazole PK/PD data are provided in Table 5.

## 8. Fluoroquinolone

Fluoroquinolones are small, lipophilic molecules, a class of antimicrobials, clinically versatile based on the broad spectrum of activity. Fluoroquinolones are active against Gram-negative bacteria such as *Enterobacteriaceae* and *Pseudomonas*, Gram-positives such as *Streptococci* and *Listeria*, and organisms without cell walls such as *chlamydia* and *mycoplasma*. They also have efficacy against mycobacterial organisms [11,169,170,171]. They are bactericidal as they inhibit the bacterial DNA replication enzymes, DNA gyrase and topoisomerase IV. Their effectiveness is further bolstered by a high degree of oral to serum bioavailability, with the less lipophilic (hence hydrophilic) ciprofloxacin reaching levels of around 70% and the more lipophilic moxifloxacin and levofloxacin reaching higher levels of 90 to 100% of absorption when administered orally. These favorable pharmacokinetic attributes extend to their penetration into CSF. Various studies have shown that moxifloxacin and, to a lesser but still effective degree, ciprofloxacin and other fluoroquinolones readily penetrate the CNS. Their CSF levels are comparable to concurrent serum levels with AUC_CSF_:AUC_serum_ ratios approaching 1.0, and this does not seem to be significantly impacted in the setting of meningeal inflammation [11,19,171]. Hence, IVT or ITT administration is not necessary. Despite these obvious advantages, the utilization of fluoroquinolones in infections of the CNS is not strongly established, though there have been investigations of their potential effectiveness in the treatment of tuberculosis-related infections [11,19,169,170,172,173]. Fluoroquinolones exhibit a PAE on Gram-negative bacteria, which allows infrequent dosing.

These attractive features seem to be complicated, but not compromised, by the tendency of fluoroquinolones to have both concentration- and time-dependent activities. Most evidence seems to suggest that the effectiveness of this antibiotic class is best characterized by measuring the ratios of AUC/MIC, as well as the Cpeak/MIC [19]. Levofloxacin was studied in patients with critical neurological conditions, who had external ventricular devices due to hydrocephalus. A levofloxacin dose of 500 mg IV every 12 h achieved high penetration into the CSF but reached concentrations that were deemed inadequate for pathogens with MIC <0.5 mg/L [174]. Hence, the authors suggested that in order to achieve higher CSF concentrations for efficacy, a higher dose needs to be administered, which would not be tolerated due to significant adverse effects. Neurotoxicity described with the use of fluoroquinolones, such as encephalopathy, seizures, peripheral neuropathy, and thought rare, worsening myasthenia gravis, are all concerns that would obviate more aggressive dosing of fluoroquinolones for CNS infections [175,176,177,178].

In direct contrast, a study of 50 healthy patients who received oral moxifloxacin for prophylaxis before urological procedures demonstrated timely and effective penetration of the antibiotic into CSF [173]. Samples of CSF were obtained over the interval of several hours after the administration of oral moxifloxacin. Those samples were incubated with isolates of penicillin-resistant *S. pneumoniae*. The study showed that moxifloxacin concentrations in CSF sampled between 2 and 6 h after oral intake had significant bactericidal activity against *S. pneumoniae*, which supports moxifloxacin being a potentially useful drug in the treatment of meningitis caused by penicillin-resistant *S. pneumoniae*. The significance of this study is the absence of meningeal inflammation. Experimental animal studies of *S. pneumoniae* and *E. coli* meningitis demonstrated good penetration into the CNS [179]. Hence, it is accordingly possible to predict even better moxifloxacin penetration of the CSF in humans with meningitis. However, standard recommendations regarding fluoroquinolone utility in bacterial meningitis are still lacking due to the absence of trials documenting its efficacy; therefore, a gap exists in the confirmatory knowledge of the potential for the clinical use of fluoroquinolones in bacterial CNS infections.

Moxifloxacin, specifically, has been more extensively investigated in the setting of tuberculous meningitis as it seems to be the fluoroquinolone with the greatest in vitro effect against tuberculosis [169]. In two small studies of one and four patients with TB meningitis, respectively, the authors were able to demonstrate that oral administration of moxifloxacin achieved a CSF AUC/MIC ratio of 56 to 132, based on the dose administered [169,170]. Notably, the latter study had patients who continued taking the medication for weeks to months with no observed adverse effects. However, the moxifloxacin effectiveness in those studies remained debatable as only oral doses of 800 mg were able to reliably achieve the AUC/MIC ratios >100 that are desired for reducing the development of drug resistance [170]. Of particular note is that the use of moxifloxacin in tuberculosis treatment has a specific caveat that the plasma concentrations and AUC of the agent may be reduced by nearly 30% if rifampin is concomitantly administered, likely due to rifampin-induced glucuronidation or sulfation [180].

### Clinical Perspectives in Consideration of PK/PD Data of Fluoroquinolones

Despite their favorable PK/PD profile as small, lipophilic molecules that enable almost complete oral absorption, and similarly complete penetration into the CSF, fluoroquinolones have very sparce supportive indications for monotherapy of CNS infections. They can be administered intravenously in combination with other antimicrobials when there is concern of concomitant Gram-negative infections, such as in the treatment of ventricular drain device infection (in combination with vancomycin for example). Another niche for their clinical utility is in tuberculous and non-tuberculous mycobacterial infections, particularly moxifloxacin. However, an area that complicates the practical use of fluoroquinolones in CNS infections is their low antimicrobial activity in CSF against *S. pneumoniae* meningitis, a common meningitis pathogen [11,19]; hence, they have no role in the treatment of *Streptococcus pneumoniae* meningitis. They have less of a PAE in meningitis as compared to other infections and should be administered twice daily. Increasing their dose as a PK means to achieve higher CSF concentration results in intolerable adverse effects. Further details on the PK/PD data relevant to moxifloxacin are provided in Table 6.

## 9. Trimethoprim (TMP)/Sulfamethoxazole

Trimethoprim (TMP)/sulfamethoxazole (SMX) (to be referred to as TMP-SMX) is a combination bactericidal antimicrobial agent. Both components of this combination work synergistically to inhibit folate synthesis in susceptible bacterial pathogens [181]. Both TMP and SMX are time-dependent bacteriostatic agents, with the potential for concentration-dependent bactericidal activity for susceptible organisms. For the treatment of meningitis, where bactericidal activity is desired, appropriate concentrations need to be achieved in the CSF [182].

TMP and SMX are small lipophilic molecules, and thus, penetration into the CSF is higher than that of beta-lactam antimicrobials or aminoglycosides in both inflammatory and non-inflammatory meningeal conditions [11]. Bishop and colleagues concluded through two studies in neurosurgical patients that CSF levels of TMP-SMX after oral administration were favorable. Specifically, a 5 mg/kg TMP and 25 mg/kg SMX IV dose achieved TMP concentrations of 0.5–3.2 mg/L and SMX concentrations of 5–40 mg/L. In a study of 15 patients without meningitis, who were administered IV TMP-SMX preoperatively at 5 mg/kg TMP and 25 mg/kg SMX, the CSF concentrations in 11 of 14 patients achieved levels exceeding MIC for *Staphylococcus aureus* and *Staphylococcus epidermidis* [177]. A similar study by Dudley and colleagues reviewed the pharmacokinetic properties of TMP-SMX regarding entry into CSF in adult patients who had normal meninges. This study used a similar dosing regimen to the Wang study, (a single IV infusion of TMP-SMX in a 1:5 ratio, 5 mg/kg of TMP, 25 mg/kg of SMX) in nine adult patients who had uninflamed meninges. According to their analysis, a loading dose based on TMP at 6 mg/kg every 8 h, or 8 mg/kg every 12 h should yield steady state peak concentrations of at least 5 mcg per mL of serum, and 160 mcg of SMX per mL of serum. The CSF penetration of TMP-SMX compared to the serum level was 18% for TMP and 12% for SMX [183]. This penetration of TMP-SMX into CSF even in non-inflamed meninges has been recommended as a rationale for the prophylaxis or therapy of CNS infections where there is minimal meningeal inflammation, such as in shunt infections [183].

TMP-SMX has desirable antimicrobial activity against common Gram-negative pathogens, such as *Enterobacter*, *Acinetobacter*, and *Serratia*, as well as Gram-positive pathogens, such as *Staphylococcus aureus* and *Listeria monocytogenes*, that can cause meningitis, and which may be only moderately susceptible or resistant to third generation cephalosporins [184]. It is important to note that TMP-SMX has a well-documented risk of causing drug-induced aseptic meningitis and is the most commonly reported antibiotic cause of this condition [185]. This risk is higher among immunocompromised patients but is also seen in immunocompetent individuals. Symptoms are identical to standard infectious meningitis, and include fever, headache, and a stiff neck, but more severe hemodynamic instability has been reported as well. Symptoms can occur hours to weeks after the initiation of TMP-SMX.

### Clinical Perspective in Consideration of TMP-SMX PK/PD Data

At doses of 20 mg/kg/day (based on TMP component) divided every 6–12 h, TMP-SMX is an agent qualified for use for CNS infections caused by susceptible bacterial pathogens. The primary indications are *Listeria monocytogenes* meningitis (as alternative therapy to ampicillin), meningitis caused by Gram-negative pathogens with reduced susceptibility to beta-lactams, such as *Enterobacter*, *Acinetobacter*, and *Serratia* species, as well as for shunt infections. ITMP-SMX also has therapeutic roles in other microbial CNS infections, such as *Toxoplasma* encephalitis, *Nocardia* CNS infections, *Stenotrophomonas maltophilia*, and some parasitic and fungal pathogens (e.g., paracoccidioidomycosis). It requires dose adjustment based on renal and hepatic functions. A concern with TMP-SMX is the potential for drug-induced aseptic meningitis, which would complicate the clinical picture of the CNS infection being treated. Hence, it is an agent that is not used empirically, and has specific therapeutic niches. Data related to PK/PD characteristics of TMP-SMX are detailed in Table 7.

## 10. Tetracyclines

Tetracyclines are a class of antimicrobials with a broad range of clinical indications. In CNS infections, they are used for suspected or confirmed neurosyphilis, Lyme borreliosis, and neurobrucellosis. *Mycoplasma pneumoniae* encephalitis is another clinical syndrome for which tetracyclines are effective [128]. They have also been described in case reports to have clinical utility in combination with other agents in the treatment of VRE meningitis [187]. For CNS infections, doxycycline is the agent with the most clinical experience and the most effective of its class, based on its favorable pharmacokinetics [188,189,190]. Its favorable PK data are namely its lipophilicity and high bioavailability after oral administration, ranging between 70 and 95% [191,192], its long elimination half-life of 12–25 h [182], and its high degree of protein binding [52]. Its availability in oral and IV formulations adds to its clinical versatility. Yim and colleagues evaluated the penetration of doxycycline into the CSF of patients with latent or neurosyphilis, treated with oral doxycycline at 200 mg twice daily for 3 weeks [193]. The mean CSF concentration was 1.3 mg/L, (serum concertation was 5.8 mg/L), which equated to a penetration that ranged between 11 and 56%, at a mean of 26%. The level achieved in the CSF was above the Treponema pallidum MIC. However, in a study by Doteval of 12 patients treated with doxycycline for suspected Lyme borreliosis, the CSF to serum level after the administration of oral doxycycline was found to vary between 8 and 35%, with a mean of 15% [194]. The difference in the CSF concentration between those studies was theorized to be due to the time of sampling of the CSF after oral administration of doxycycline. In this study, the higher dose of 200 mg bid was found to achieve a CSF therapeutic level more rapidly than the 100 mg bid dose, and accordingly endorsed by the authors as preferable in cases of mild neuroborreliosis in outpatients. Hence, doxycycline is the preferred tetracycline for neuroborreliosis, and the preferred dosing is 200 mg every 12 h, orally for outpatients, or intravenously in the appropriate clinical setting.

Tigecycline is a newer tetracycline that is approved for community-acquired pneumonia, skin and soft tissue infections, and complicated intraabdominal infections. It is available only in IV formulation. It is a derivative of minocycline and has the ability to resist efflux from bacterial cells and avoid mechanisms of bacterial ribosomal protection. It is active against many MDROs including MRSA, MDR *Acinetobacter baumannii*, and carbapenemase-producing *Enterobacteriaceae*, but has no antimicrobial activity against *Pseudomonas aeruginosa* [195,196,197].

Tigecycline has been associated with increased mortality in several studies [198,199,200], and is not recommended as monotherapy [201]. Reasons have been postulated to be due to low serum levels resulting in a suboptimal AUC/MIC [202]. However, higher doses than the recommended 100 mg loading dose, then 50 mg IV every 12 h, have been associated with a higher frequency of adverse effects, particularly gastrointestinal, and the safety of such a regimen is not known [203,204,205]. Hence, using this pharmacodynamic approach to augment serum concentrations in an attempt to improve CNS levels is not an option.

Tigecycline crosses the BBB less efficiently than doxycycline, with a low CSF concentration of 0.11 mg/dL [206]. Hence, IVT or ITT administration becomes an attractive option and has been reported by several investigators in doses ranging from 2 to 10 mg twice daily. These data are published in several case reports describing the use of ITT and IVT tigecycline for the treatment of extensively drug-resistant *Acinetobacter baumannii* or MDR *Klebsiella* infections of the CNS with favorable outcomes [32,33,35,36,37,207]. Combination with colistin, both administered intraventricularly, has also been described in small series or case reports for similar organisms [208,209,210,211]. Other studied regimens in an individual patient were a combination of IV/IVT tigecycline–amikacin for carbapenem-resistant *Klebsiella pneumoniae* ventriculitis [212]. For highly resistant enterococcal infections, IV tigecycline was reported in combination with IVT daptomycin in a toddler [213], and both IV plus IVT tigecycline for daptomycin-resistant VRE in an infant [214]. Another potential niche for therapy is in rickettsial CNS infections when IV doxycycline is not available and oral doxycycline was not tolerated [215].

Despite the above series of cases delineating no directly attributable side effects to IT/IVT administration of tigecycline, a recent report by Li et al. described a case of spinal arachnoiditis after ITT tigecycline treatment for XDR *Acinetobacter baumannii* CNS infection related to a ventriculoperitoneal shunt [34]. This complication occurred after nine doses of ITT tigecycline and resolved after its discontinuation. With increasing use of IT/IVT tigecycline for MDR CNS infections, more of such reports will surface, and, hence, need to be monitored in the literature.

### Clinical Perspective in Consideration of Doxycyline and Tigecycline PK/PD Data

Doxycycline is utilized as an alternative agent to penicillin or cephalosporins for neuroborreliosis, with a recommended dose of 200 mg every 12 h, orally for outpatients, or intravenously in the appropriate clinical setting. As for tigecycline, the newer IV tetracycline, has a narrower niche for CNS infections. With the variability in data regarding the appropriate dosing and frequency of ITT administration of tigecycline, as well as the absence of its evaluation in larger numbers of patients, the use of IT/IVT tigecycline is to be reserved almost exclusively for cases with MDR or XDR organisms, especially *Acinetobacter baumannii* or CRE *Klebsiella*, for which other antimicrobial agents are not available or have failed. There is no role for IV tigecycline for the treatment of CNS infection due to poor penetration of the BBB. Further details regarding the PK/PD data relevant to doxycycline and tigecycline are summarized in Table 8.

## 11. Polymyxin B and Colistin

Polymyxins are a class of antimicrobials increasingly being utilized as a last-line therapeutic option for multidrug-resistant Gram-negative bacteria. Polymyxin B has rapid bactericidal activity against multidrug-resistant pathogens, such as *Pseudomonas aeruginosa*, *Acinetobacter baumannii*, *and Klebsiella pneumoniae*. Colistin is polymyxin E, a polypeptide effective in the treatment of resistant Gram-negative organisms. It is notable that cross-resistance does exist between polymyxin B and colistin. The chemical structure of polymyxins includes a mixture of lipophilic and hydrophilic groups, which is essential to their mechanism of action, allowing them to penetrate the outer membrane of Gram-negative pathogens. Resistance among polymyxins generally involves the expression of outer membrane proteins, such as efflux pumps [216]. Studies examining the pharmacodynamics of polymyxin show concentration-dependent bactericidal activity against *Pseudomonas, Klebsiella, and Acinetobacter* [217]. The same authors indicate that regrowth could occur at colistin concentrations up to 64 x MIC, and report that population analysis profiles (PAPs) demonstrate the existence of a small proportion of colistin-resistant strains. This suggests that polymyxin monotherapy may promptly lead to the selection of bacterial resistance.

Current studies suggest that the penetration of polymyxins through the BBB is variable. Chen and colleagues reported a study of 28 neurosurgery patients who developed intracranial infection with multidrug-resistant organisms and were treated with polymyxin B and ventricular drainage. The results demonstrated bacterial clearance from CSF at 92.9% and a clinical cure rate at 82.1% [218].

A small study investigating colistin in five critically ill adult patients showed a CSF:serum ratio of 0.051 to 0.057 mcg/mL with 5% penetration [28]. Bergen and colleagues similarly reported variations in CSF:serum ratios of 0.051 to 0.057, corresponding to a concentration in CSF ranging from 0.041 to 0.099 mcg/mL [217]. These reports suggest that polymyxins are not suitable for meningitis treatment, as concentrations fall below MIC breakpoints. Information from studies on colistin are further complicated by the lack of differentiation between the sodium salt form of colistin (colistin methanesulphonate) and standard colistin.

A tertiary study of case reports and case series from 1950–2006 reviewed meningitis interventions comprised of monotherapy or combined therapy with IV or ITT polymyxin B or colistin, often used after the failure of prior antimicrobial treatment, and examined the efficacy of polymyxin treatment. Thirty-one studies were included in the report with 60 patients experiencing 64 episodes of bacterial meningitis treated with polymyxin-containing treatment regimens. Polymyxin monotherapy was utilized in 56% of Gram-negative meningitis cases with doses ranging from 20,000 to 250,000 IU in adults, and 5000 to 120,000 IU in pediatrics. The duration of treatment was 1–9 weeks. The total outcome was an 80% cure rate (51 of 64 episodes). Toxicity was reported in 28% of cases, with meningeal irritation being the most common adverse effect [219].

Recent reports have demonstrated that infections with carbapenemase-producing *Klebsiella pneumoniae*, an infection with a high mortality rate, have had high cure rates when colistimethate sodium was combined with tigecycline or rifampin. Synergistic bactericidal activity with these combinations has been shown to reach appropriate concentrations [48]. Additionally, IVT colistin, both as an adjunctive and alternative therapy, achieved higher CSF concentrations and 100-fold increased AUCs than with IV doses. IV doses were unable to reach concentrations above 2.75 mg/L in CSF [24].

### Clinical Perspectives in Consideration of Polymyxin PK/PD Data

Colistin as a prototype of this class has limited therapeutic roles in CNS infections. Polymyxins are utilized as alternatives in infections with MDRO Gram-negative bacterial species such as *P. aeruginosa, Acinetobacter*, *and Klebsiella pneumoniae* when there are very limited options for the use of other antimicrobial drugs. Although the availability of colistimethate sodium has made this class more tolerable, it still has significant toxicity profiles, and the penetration into CSF is limited, which has led to the evaluation of IVT administration [48]. Its primary utilization is in the treatment of MDRO Gram-negative rods, particularly *Acinetobacter baumannii*, and should be utilized with other antimicrobials in the treatment of MDRO CNS infections. Its clinical utility is primarily in ITT or IVT administration, as the IV route does not achieve a CSF blood level that would effectively treat CNS infections [28]. Details of PK/PD data related to polymyxin B and colistin are summarized in Table 9 and Table 10.

## 12. Conclusions

This review summarizes the literature of the optimal use of antimicrobials in the management of CNS infections, from the perspective of their PK/PD data. It focuses on the parameters that optimize the dose, route of administration, and drug-related characteristics. Optimal utilization of antimicrobial therapy occurs only with good knowledge of PK/PD metrics related to the individual drug being utilized. PD principles of concentration- vs. time-dependent activity should be applied when using drugs for CNS infections, as this allows innovative dosing and schedules, while knowledge of PK governs routes such as oral, IV, or alternatives such as IVT or ITT administration. The BBB (and BCSFB) functions primarily as the controller of the preferential passage of certain molecules vs. others, and that function is disrupted with the inflammation that occurs with meningitis or ventriculitis. For example, low BBB permeability prevents beta-lactams from passing freely, although that same BBB exhibits preferential allowance of cephalosporin passage, which explains the huge role of ceftriaxone, ceftazidime, and cefepime in today’s therapeutic regimens. Alternatively, although carbapenems cross to a lesser degree than cephalosporins, meropenem has a preferential role in several infections as it achieves a high fT > MIC in CSF. Vancomycin, another cornerstone antimicrobial for CNS infections, achieves concentrations in CSF that are at least partially dependent on the level of meningeal inflammation with relatively low CNS penetration overall and variable CNS concentrations following systemic dosing alone. That drives the need to ensure appropriate dosing and monitoring of vancomycin systemically, to improve CSF penetration and ascertain CSF levels above the MIC of the organism being treated. Other antibacterial classes, such as metronidazole, linezolid, and fluoroquinolones, have significantly better CSF penetration than beta-lactams and glycopeptides, with a documented CSF/plasma ratio above 80%, which is further bolstered with meningeal inflammation. Other antimicrobials that poorly penetrate into the CSF may be the sole option available to treat certain MDROs. Drugs, such as tigecycline, colistin, and daptomycin, achieve optimal delivery into the CSF by direct routes, such as ITT or IVT.

One must remember that interpatient variability is prominent and often explains the variable responses such that some patients do better than others when exposed to the same therapeutic approach for the same clinical condition being treated. As the armamentarium of available antimicrobials broadens, it becomes more critical to recognize the best available first-line drugs, their alternatives, and those used for de-escalation while maintaining clinical efficacy and simultaneously avoiding toxicity and preserving patient safety.

With the broader availability of antimicrobial choices, the parallel increase in antimicrobial resistance, together with the risks of severe infections due to immune senescence or immune compromise beg for maximizing the evaluation of available newer agents in the management of CNS infections so as to expand therapeutic options. There remain many gaps in our knowledge of the optimal strategies in the management of CNS infection. Examples of those are the synergism of combination therapies, use of corticosteroids or other immune modulatory agents to enhance the effectiveness of antimicrobial therapy, achievable concentrations at the site of infection, optimal delivery mechanisms, toxicity of newer agents that are rarely utilized, and dose optimization as pertains to the patient’s clinical status. These should be the next phase of research in the management of CNS infections.

## Figures and Tables

**Figure 1 antibiotics-11-01843-f001:**
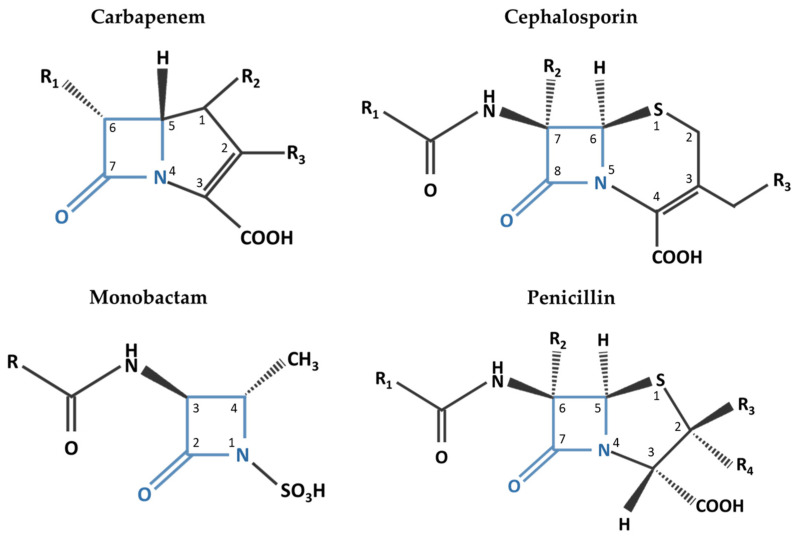
Molecular Structure of the Beta-Lactams.

**Figure 2 antibiotics-11-01843-f002:**
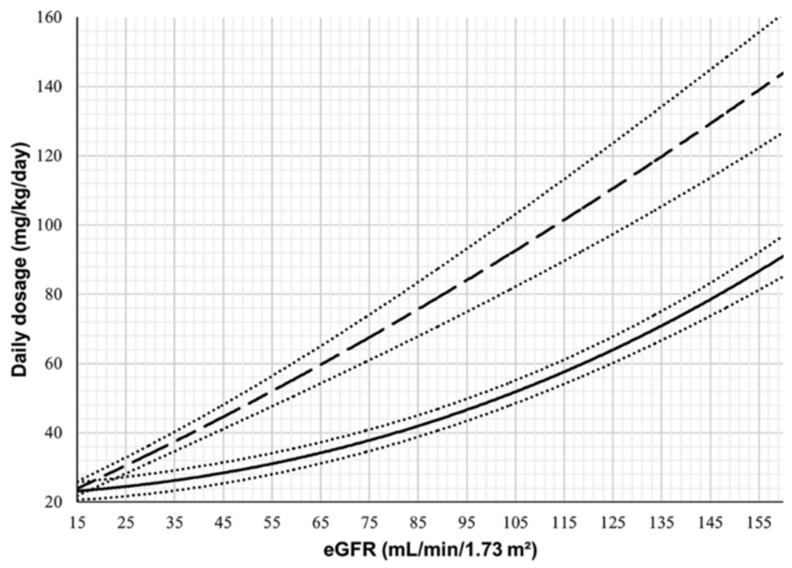
Nomogram of the daily dose of ceftriaxone per kilogram of total weight to be administered to achieve a trough concentration target of 20 mg/L (full line) and to not exceed 100 mg/L (broken line) with a probability of 0.9, accounting for renal function estimated by the CKD-EPI formula (eGFR) using a twice-daily regimen. Dotted lines represent the 95% confidence interval (used with permission from Antimicrobial Agents and Chemotherapy, License ID1286030-1, ISSN1098-6596) [113].

**Table 2 antibiotics-11-01843-t002:** Vancomycin dosing and PK/PD data.

Vancomycin Dose Requires Renal Dose Adjustment	IV: 30–60 mg/kg/Day ITT: 5–20 mg Daily IVT: 5–20 mg Daily
Indication/targeted organisms	Gram-positive organisms
PK/PD data	CSF/serum concentrations: ▪ Uninflamed meninges: 0 to 4 mg/mL ▪ Inflamed meninges: 6 to 11 mg/mL, ratio 80% Percent protein binding: 30–60% Serum half-life: 2–7 h Cmax: Variable AUC/MIC: >400

References [48,121,122,127].

**Table 3 antibiotics-11-01843-t003:** Linezolid dosing and PK/PD data.

Linezolid Dose	IV/PO: 600 mg Twice Daily
Indication/targeted organisms	Vancomycin-resistant Enterococcus (VRE), methicillin-resistant *Staphylococcus aureus* (MRSA), and *Propionibacterium acnes* CNS infections
PK/PD data	CSF/serum concentrations: 66–77% Percent protein binding: 31% Serum half-life: 2–10 h Cmax: 18–23 mg/L

References [133,137,146].

**Table 4 antibiotics-11-01843-t004:** Daptomycin dosing and PK/PD data.

Daptomycin Dose Requires Weight and Renal Dose Adjustment	IV: 6–10 mg/kg Once Daily IVT: 5 mg Daily or Every 48 h
Indication/targeted organisms	Vancomycin-resistant *Enterococcus* (VRE) and methicillin-resistant *Staphylococcus aureus* (MRSA) CNS infections
PK/PD data	CSF/serum concentrations: 0.45% Percent protein binding: >90% Serum half-life1: 4–9 h Cmax: 0.24%

References [27,29,151].

**Table 5 antibiotics-11-01843-t005:** Metronidazole dosing and PK/PD data.

Metronidazole dose Does not require renal adjustment; hepatic adjustment to 50% dose in severe impairment	Orally or intravenously (500 mg over 30 min every 8 h)
Indications/targeted organisms:	Anaerobic bacteria (*Bacteroides fragilis, Prevotella species, Fusobacterium necrophorum, Clostridium difficile, Gardneralla vaginalis*), protozoa, and microaerophilic bacteria.
PK/PD data	Serum/CSF Penetration: 18–103% CSF/Serum AUC ratio: 0.86–1.02 Serum Cmax: 6.2–40.6 mg/L CSF Cmax: 11.0–84.1 mg/L Protein Binding: <20% Elimination half-life: 3.1–16.4 h

References [48,162,164,165,166,167,168].

**Table 6 antibiotics-11-01843-t006:** Moxifloxacin dosing and PK/PD data.

Moxifloxacin Dose Requires no renal dose adjustment	IV or PO 400 mg daily (except possibly when co-administered with rifampin, then consider 800 mg daily)	
Indication/targeted organisms	Tuberculous Meningitis	
PK/PD Data for moxifloxacin	Serum/CSF penetration: Ratios ranged from 0.0913 to 0.741, depending on time after administration Peak ratio at 4–6 h CSF/serum AUC ratio: Uninflamed/mildly inflamed meninges: 0.45 Strongly inflamed meninges: 0.79 (0.79–0.94) Serum Cmax: Moxifloxacin 400 mg/day: 4.5 mg/L Moxifloxacin 800 mg/day: 2.45–3.65 mg/L Protein binding Moxifloxacin 400 mg/day: 50–60% in serum, 10% in CSF Moxifloxacin 800 mg/day: 40% in serum, 5% in CSF Elimination half-life: Moxifloxacin 400 mg/day: 4.55–12 h (5.52–6 h in CSF) Moxifloxacin 800 mg/day: 4.09 h (5.20 h CSF)	

References [11,169,170,171,173]

**Table 7 antibiotics-11-01843-t007:** Sulfamethoxazole/Trimethoprim dosing by indication/targeted organisms and PK/PD data.

Intracranial/spinal epidural abscess (MRSA)	IV: 5 mg/kg/dose every 8–12 h
Melioidosis (*Burkholderia pseudomallei*)	Oral/IV (40–60 kg): 240 mg twice daily Oral/IV (>60 kg): 320 mg twice daily
Meningitis (MRSA, *Listeria monocytogenes*, *E. coli*, *Enterobacteriaceae*)	IV: 5 mg/kg/dose every 6–12 h
Nocardiosis (off-label use, not recommended for monotherapy)	IV: 15 mg/kg/day divided into 3–4 doses
*Toxoplasma gondii* encephalitis	● Prophylaxis ○ Oral: 1 double strength tablet once daily ● Secondary prophylaxis ○ Oral: 1 double strength tablet twice daily ● Treatment ○ Oral/IV: 10 mg/kg/day divided into 2 doses
PK/PD Data	CSF/serum concentrations: TMP: 0.23–0.53 SMX: 0.20–0.36 Percent protein binding: TMP: ~44% SMX: ~70% Serum half-life: TMP: 6–11 h SMX: 9–12 h Cmax: CSF (TMP): 1 mcg/mL CSF (SMX): 13.8 mcg/mL Serum (TMP): 5 mcg/mL Serum (SMX): 160 mcg/mL

References [183,186].

**Table 8 antibiotics-11-01843-t008:** Tetracyclines indications, dosing, and PK/PD data.

	Doxycycline	Tigecycline
Indication	Neurosyphilis (alternative)	Meningitis with MDR or XDR organisms (*Acinetobacter baumannii* or CRE *Klebsiella*)
Dose Requires no renal or hepatic dose adjustment	IV 200 mg every 12 h PO: 200 mg every 12 h No IVT/ITT administration	Only IVT or ITT: Dosing range: 2 to 10 mg twice daily No role for IV therapy (except in combination with IVT/ITT, case reports. See text.
PK/PD Data	Oral bioavailability: 70–95% Elimination half-life: 12–25 h Protein binding: 93% Serum to CSF penetration: mean 26%; range 11v56%, based on a dose of 200 mg every 12 h	

References [14,15,52,188].

**Table 9 antibiotics-11-01843-t009:** Polymyxin B dosing and PK/PD data.

Dose	IV/IVT: 50,000 units once daily (in combination with systemic therapy)
Indication/targeted organisms	CSF Shunt-related meningitis (MDR *Pseudomonas aeruginosa*, *Acinetobacter baumannii*, *Klebsiella pneumoniae*)
PD/PD Data	CSF/serum concentrations: No data available Percent protein binding: 58% Serum half-life: 9–11.5 h Cmax: 2–14 mcg/mL

References [220,221].

**Table 10 antibiotics-11-01843-t010:** Colistin dosing and PK/PD data.

Dose Requires weight and renal dose adjustment	IV/IVT: 10 mg once daily (Colistimethate sodium)
Indication/targeted organisms	Meningitis (MDR *Pseudomonas aeruginosa*, *Acinetobacter baumannii*, *Klebsiella pneumoniae*)
PK/PD Data	CSF/serum concentrations: 0.05 Percent protein binding: No data available Serum half-life: 251 min Cmax: no data available

References [28,220,221].

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
