# Peer review of "The Blood–Brain Barrier and Pharmacokinetic/Pharmacodynamic Optimization of Antibiotics for the Treatment of Central Nervous System Infections in Adults"

_antibiotics, 2022, doi:10.3390/antibiotics11121843_

Round 1

Reviewer 1 Report

This paper aimed to review the pharmacokinetics/pharmacodynamics of antibiotics in the aspect of BBB penetration and treatment of CNS infections. Some specific points are detailed below.

1.     The title

It is recommended to revise the title to be more specific in order to better represent the contents of the manuscript. The main content of the manuscript could not be grasped from the current title.

2.     Tables

-       As summarized for the beta-lactam antibiotics in Table 1, it would be good to display the PK data (CSF/Serum ratio, etc.) of other groups of antibiotics in the table as well.

Author Response

Thank you for this constructive feedback. We will be happy to proceed with suggested reviews. As first author, following are my responses:

  1. This is a very insightful remark. Our title will be further fine-tuned to be: "The Blood Brain Barrier and Pharmacokinetic/Pharmacodynamic Optimization of Antibiotics in the Treatment of Central Nervous System Infections in Adults".
  2. Will do that. I would like to request 5 additional days of extension, to resubmit with the updated information in each table by Dec. 5, 2022.

Reviewer 2 Report

In this article by Haddad N and Colleagues, the authors have nicely and adequately reviewed recent literature on the use of antibiotics to treat bacterial CNS infections.

The article is well organized and provide a complete overview for each class of antibiotics used to treat CNS infections.

I have only minor comments that should be addressed: 

1) according to the title, the authors should add more considerations about the PK/PD properties for each discussed antibiotic.

Perhaps, they could enrich tables at the end of each antibiotic's paragraph by including more info such as PK/PD ratio, PK parameters (% protein binding, half life and if possible Cmax). Similarly, PK parameters already described within text could be moved to the tables.

Author Response

Thank you very much for the remarks and suggestions. We shall accordingly update the tables to include the required information. I do believe they are  appropriate comments. For that, I am requesting from the editors an extension of 5 days.